# A Depression Diagnosis Method Based on the Hybrid Neural Network and Attention Mechanism

**DOI:** 10.3390/brainsci12070834

**Published:** 2022-06-26

**Authors:** Zhuozheng Wang, Zhuo Ma, Wei Liu, Zhefeng An, Fubiao Huang

**Affiliations:** 1Faculty of Information Technology, Beijing University of Technology, Beijing 100124, China; mazhuo@emails.bjut.edu.cn (Z.M.); liuwei0823@bjut.edu.cn (W.L.); 2Advising Center for Student Development, Beijing University of Technology, Beijing 100124, China; anzf@bjut.edu.cn; 3Department of Occupational Therapy, China Rehabilitation Research Center, Beijing 100068, China; huangfubiao@sina.com

**Keywords:** depression, electroencephalogram (EEG), one-dimensional convolutional neural network (1D-CNN), gated recurrent unit (GRU), attention mechanism

## Abstract

Depression is a common but easily misdiagnosed disease when using a self-assessment scale. Electroencephalograms (EEGs) provide an important reference and objective basis for the identification and diagnosis of depression. In order to improve the accuracy of the diagnosis of depression by using mainstream algorithms, a high-performance hybrid neural network depression detection method is proposed in this paper combined with deep learning technology. Firstly, a concatenating one-dimensional convolutional neural network (1D-CNN) and gated recurrent unit (GRU) are employed to extract the local features and to determine the global features of the EEG signal. Secondly, the attention mechanism is introduced to form the hybrid neural network. The attention mechanism assigns different weights to the multi-dimensional features extracted by the network, so as to screen out more representative features, which can reduce the computational complexity of the network and save the training time of the model while ensuring high precision. Moreover, dropout is applied to accelerate network training and address the over-fitting problem. Experiments reveal that the 1D-CNN-GRU-ATTN model has more effectiveness and a better generalization ability compared with traditional algorithms. The accuracy of the proposed method in this paper reaches 99.33% in a public dataset and 97.98% in a private dataset, respectively.

## 1. Introduction

Depression is a common and serious mental illness. According to WHO, there are more than 350 million people worldwide suffering from depression [1]. Among them, two-thirds of the patients have had thoughts of suicide, and more than half of the patients have tried self-harm behavior. It is estimated that by 2030, depression will be the largest burden of disease in the world [2]. In addition, the incidence of depression is beginning to show a younger trend. According to surveys and studies, nearly 30 million children and adolescents have suffered or are suffering from depression [3]. If the screening and diagnosis of depression can be conducted in the early stage, psychological or drug treatment can be carried out in time for patients. However, due to insufficient medical resources in the field of mental health, the recognition rate of mental health diseases such as depression is only 21%. Nearly 80% of patients with depression have not been discovered [4]. The main reason for this is the lack of an objective evaluation index and efficient detection method. In most cases, doctors ask patients some questions in combination with scales, such as the Self-Rating Depression Scale (SDS) [5] and Beck Depression Inventory (BDI) [6], but this method lacks objectivity. For example, the patient may not answer the doctor’s questions truthfully in order to conceal the condition, or the patient’s limited education level may not allow them to accurately describe their own situation, which may lead to misdiagnosis. Depression seriously affects people’s work, study, and life. Therefore, it is necessary to find an objective and accurate diagnosis method for depression.

Electroencephalograms (EEGs) have been explored as an effective biomarker and diagnosis tool for the detection of neurological disorders such as depression, epilepsy, seizure, Alzheimer’s, and Parkinson’s and in the analysis of emotions in comparison to other tools because of their non-invasive and economical nature [7,8,9]. EEG signals are the overall response of the electrophysiological activity of human brain nerve cells in the cerebral cortex or scalp surface [10]. Clinical studies have shown that EEG signals contain a lot of physiological and pathological information [11], and different mental states will affect the changes of EEGs, which provides an important basis for the diagnosis of depression and other brain diseases. Campisi, P. et al. [12] pointed out that the brain regions of patients with depression have changed, which reduces the decision-making ability of patients with depression. Koller-Schlaud et al. [13] discovered that, in the resting state, theta activity at the central electrode site is highly distinct, and they were able to discriminate between healthy controls and bipolar depressed controls using an analysis of the variance model. Kang et al. [14] found that the greatest performance for the classification model for diagnosing depression was achieved using alpha asymmetry images. For depression patients, Liu et al. [15] found a substantial link between the long-distance edge of the beta band, which was distributed largely within the frontal brain areas and between the frontal and parietal–occipital brain areas.

According to the frequency property of EEG signals, we can complete the diagnosis and treatment of neurological diseases by detecting and recording human EEG signals. In order to distinguish patients with depression from normal people, Erguzel et al. [16] used a genetic algorithm to select the multidimensional features of EEG signals, and then input the selected features into a back propagation neural network classifier. The accuracy of this method was 89.12%. Hosseinifard et al. [17] analyzed the effectiveness of extracting non-linear features from the EEGs of depression patients and normal subjects. The study performed feature selection on the extracted non-linear features, and then input them into the logistic regression classifier for the classification of normal and depression categories. Mumtaz et al. [18] used a support vector machine classifier to distinguish between normal and depressed EEG signals and extract specific clinical features from EEG signals. Liao et al. [19] used the public space pattern of the nuclear feature filter bank and principal component analysis to extract features from EEG signals for classification. They reported a diagnostic accuracy of 81.23%. The authors of [20] calculated the fuzzy entropy of 16-channel EEG signals as the feature, and designed a fuzzy function based on the neural network as the classifier based on the existing neural network, which achieved an accuracy of 87.5% in the classification task of patients with severe depression and normal people.

In recent years, in biomedical signal processing, methods based on deep learning have replaced traditional machine learning methods. Compared with traditional machine learning, deep learning includes multi-layer hidden structures that can automatically extract effective features [21]. In addition, deep learning can better approximate complex functions and effectively process high-dimensional and nonlinear data, so as to avoid the problem of insufficient diagnostic capabilities. In 2018, Acharya et al. [22] used a deep learning method to detect depression with a 13-layer CNN and obtained a high accuracy. In 2019, Nagabushanam et al. [23] proposed a two-layer long short-term memory (LSTM) network to extract statistical features from the input EEG signal collected from the database, namely the mean, standard deviation, kurtosis, and skewness. Then, the features were applied to the LSTM classification model, and a better performance than that of traditional methods was achieved. In the same year, Ozal et al. [24] proposed a deep hybrid model combining CNN and LSTM to detect depression through EEG signals. The experiment achieved a high accuracy in both the left and right brains.

In this paper, a method for depression prediction based on CNN, GRU, and the attention mechanism is proposed to improve the accuracy of depression diagnosis.

## 2. Materials and Methods

### 2.1. Data Sources and Data Preprocessing

There were two datasets used in this study: (1) the public dataset for depression analysis provided by Lanzhou University (MODMA dataset) [25] and (2) a private dataset.

MODMA dataset

A total of 53 participants were included in the experiment, including 24 outpatients (13 men and 11 women, 16–56 years old) diagnosed with depression and 29 healthy controls (20 men and 9 women, 18–55 years old). The experiment used a 128-channel HydroCel geodetic sensor network and Net Station acquisition software to record continuous EEG signals. The sampling frequency was 250 Hz. The reference electrode was Cz. The experiment recorded the EEG signals of the resting state with eyes closed for 5 min. Participants were asked to remain awake and stationary without any physical movement, including head or legs, as well as any unnecessary eye movements, such as eye jumps and blinks. For the consideration of the time performance and calculation efficiency, we selected 16 electrodes among them (Fp1/2, F3/4, C3/4, P3/4, O1/2, F7/8, T3/4, T5/6) instead of 128 electrodes. Depression diagnosis using these electrodes has also been demonstrated in previous studies [26].

Private dataset

There were a total of 32 participants in the experiment, of which 16 patients with depression (7 men and 9 women, 18–35 years old) were confirmed cases from a tertiary hospital in China and the 16 healthy controls (9 men and 7 women, 18–26 years old) were college students who filled out the SDS before the experiment, and the scores were all below 30. The experiment used OpenBCI, a relatively common physiological signal acquisition platform in the world. A 16-channel EEG cap was used in the experiment. The positions of the electrodes were placed in accordance with the international standard 10–20 system, with Cz as the reference electrode, as shown in Figure 1 [27]. The resting-state EEG signal of each subject was collected for 3 min, and the sampling frequency of the signal was 250 Hz. All subjects signed an informed consent form before the experiment and were paid after the experiment was completed. Table 1 describes the two datasets in detail.

In EEG research, data preprocessing is an indispensable step, because the EEG signal is very weak, and the noise and interference in the collected signal are often stronger than the real EEG signal. Subsequent series of analyses may have large errors. After preprocessing, a relatively ideal high-quality EEG signal can be obtained, and subsequent analysis can be performed on this basis. Figure 2 shows the flow of EEG signal preprocessing.

For the EEG signal, the most obvious interference is the 50 Hz/60 Hz power frequency signal (50 Hz in China). Therefore, firstly, a 50 Hz notch filter was selected to filter the signal for the first time to remove the interference caused by the power frequency. The EEG signal is mainly concentrated between 0.5 and 50 Hz, so the IIR or FIR filter can be applied as the band-pass filter. In electrophysiology, the Butterworth or Chebyshev IIR filter is commonly used; the design of FIR is complex, and it is difficult to set a reasonable order for EEG signal analysis. The Butterworth filters have no passband and stopband ripple, having the shallowest roll-off near the cutoff frequency compared to the other commonly used Chebyshev and elliptic IIR filters [28]. In this study, considering the time performance for real-time signal calculation, the fourth-order Butterworth band-pass filter was selected to filter the signal in the second stage to remove the baseline drift of low frequency, electrocardiogram (ECG) interference, and other high-frequency noise.

Since the EEG signals and the electrooculogram (EOG) signals and electromyogram (EMG) signals overlap in the frequency band, only two filterings cannot completely remove the interference of the EOG and EMG. In this study, fast independent component analysis (Fast ICA) [29] was applied to remove EOG and EMG artifacts [30]. The Fast ICA algorithm is derived from the cocktail party problem. It can effectively solve the problem of blind source separation by estimating the linear equations. All independent components in the EEG signal are regarded as “signal sources”, and the original signal collected by the device is regarded as “microphones”. The core of the Fast ICA algorithm is to find a matrix that can be de-mixed to make the new signal after de-mixing as close to the original signal as possible.

In this paper, the MNE library in Python was used to remove the artifacts, and each component is visualized. The independent components of the EOG artifacts have typical characteristics, that is, front-end distribution and high-amplitude spikes, so the EOG artifacts can be easily removed. After obtaining clean EEG signals, the EEG signal data of all subjects were cut into 1 s time segments. The purpose of this operation was to increase the number of samples that could be used in the experiment to make the experiment more convincing.

Power spectral density (PSD) [31] is an important indicator of resting-state EEG signals, and PSD can reflect the energy of EEG signals in different bands and channels. Studies have shown that there are differences in certain bands between the EEG signals of depressed patients and normal people [32,33]. Therefore, the PSD of EEG signals was selected as the feature to identify patients with depression and normal people. This paper uses Welch’s method to extract the power spectrum features of EEG signals. For the EEG signal x(n) of a certain channel, it is first divided into L small fragments. Each small fragment has M sampling points, then the ith small fragment $x_{i}\left(n\right)$ can be recorded as


(1)
$$x_{i}\left(n\right)=x\left(m+iM-M\right),0\leq m\leq M,1\leq i\leq L$$


After that, each segment of data is windowed. In this study, a Hamming window w(n) with a length of 100 is applied, because the main lobe of the rectangular window is narrow, and the frequency resolution is high. After windowing, the periodogram $I_{i}\left(\omega \right)$ of each segment of data is obtained, where the periodogram of the *i*th segment is [31]
(2)$$I_{i}\left(\omega \right)=\frac{1}{U}{\left|\sum_{n=0}^{M-1}x_{i}\left(n\right)w\left(n\right)e^{-j\omega n}\right|}^{2},$$
where (3)$$w\left(n\right)=\left\{\begin{array}{cc} 1, & 0\leq n\leq 99\\ 0, & otherwise\; \end{array}\right.$$
(4)$$U=\frac{1}{M}\sum_{n=0}^{M-1}w^{2}\left(n\right),$$

Then, the power spectral density is [34](5)$$P_{xx}\left(e^{j\omega }\right)=\frac{1}{L}\sum_{i=1}^{L}I_{i}\left(\omega \right),$$

This paper uses the above calculation method to extract the PSD features of 5 bands of 16 channels in the EEG signals as the input samples of the depression detection model.

### 2.2. Model Preparation

In this part, the components of the proposed network model are introduced, respectively, and then the proposed network structure is described.

#### 2.2.1. CNN

Convolutional Neural Network (CNN) was first proposed by LeCun et al. and achieved success in the recognition of handwritten digits [35]. CNN is a feed-forward neural network, which has a deep structure of convolution calculation. It is one of the representative algorithms of deep learning. CNN can reduce the number of parameters by local receptive fields and shared weights to avoid over-fitting. The typical CNN architecture is shown in Figure 3.

The convolution kernel of the convolution layer can fully extract features, and the pooling layer can compress features, reducing the number of training parameters. The convolution process of a 1D-CNN is shown in Figure 4.

Suppose $X=\left[x_{1},x_{2},\cdots ,x_{n}\right]\in R^{m\times d}$ is the input sequence data of the network, in which *n* indicates the length of the input sequence, and *d* represents the dimension of the sequence. The essence of the convolution layer is that multiple filters (or convolution kernel) are convoluted with the input. The convolution operation is defined as Equation (6):
(6)$$c_{i}=\varphi \left(w\cdot x_{i:i+m-1}+b\right)$$ where $w\in R^{m\times d}$ represents the convolution kernel, $x_{i:i+m-1}$ represents a sliding window of length m, φ represents the network activation function, and b represents the offset parameter.

#### 2.2.2. GRU

Recurrent Neural Network (RNN) is a network that takes sequence data as input, recursively in the evolution direction of the sequence, and all nodes are connected in a chain [36]. However, the RNN has a long-term dependencies problem, that is, when learning a long sequence, the RNN will have gradient vanishing or gradient explosion phenomena, which makes it impossible to master the long-term non-linear relationship of span [37]. In order to solve the above problems, Hochreiter and Schmidhuber proposed Long Short-Term Memory (LSTM) network in 1997 [38]. In 2014, K. Cho proposed the Gated Recurrent Unit (GRU) based on LSTM [39]. It combines the forget gate and the input gate in LSTM into an update gate. In addition, it also has a reset gate. The update gate defines the amount of the previous memory saved to the current time step, and the reset gate determines how to combine the new input information with the previous memory. The update gate, reset gate, output candidate of GRU, and output of the current hidden layer are calculated using the following equations [40]:(7)$$z_{t}=\sigma \left(W_{xz}x_{t}+W_{hz}h_{t-1}+b_{z}\right)\;\;\;\;t=1,2,\ldots ,T,$$
(8)$$r_{t}=\sigma \left(W_{xr}x_{t}+W_{hr}h_{t-1}+b_{r}\right)\;\;\;\;t=1,2,\ldots ,T,$$
(9)$${\tilde{h}}_{t}=tanh\left(W_{x}x_{t}+W_{h}\left(h_{t-1}*r_{t}\right)+b\right)\;\;\;\;t=1,2,\ldots T,$$
(10)$$h_{t}=\left(1-z_{t}\right)*h_{t-1}+z_{t}*{\tilde{h}}_{t}\;\;\;\;t=1,2,\ldots ,T,$$
where σ represents the sigmoid activation function, W denotes recursive weight matrices, b denotes bias vectors, and subscripts z, r, and h represent the update gate, reset gate, and output of the current hidden layer, respectively.

#### 2.2.3. Attention Mechanism

The attention mechanism originated from the study of human visual function and was initially used for machine translation [41]. In recent years, it has been widely used in different deep learning tasks such as natural language processing, image recognition, and speech recognition [42]. In a classification task, not all features contribute equally to classification. The attention mechanism devotes more attention resources to the target area that needs to be focused on to obtain more detailed information about the target, and to suppress other useless information.

The attention mechanism in deep learning is to assign different weights to inputs or features. Important information is assigned more weight, and relatively unimportant information is assigned less weight. The workflow of the attention mechanism is first to score the hidden state of each unit in the encoder, then use the softmax function to normalize the scores to obtain the weight. Finally, the weighted summation of the hidden states of all time steps is performed to obtain the input variables of the next layer.

The general expression of the attention mechanism is [43]
(11)$$O=\mathrm{softmax}\left(QK^{T}\right)\cdot V$$
where *Q* is the query item matrix, *K* is the corresponding key item, and *V* is the value item to be weighted average.

#### 2.2.4. Proposed Depression Diagnosis Approach

CNN is widely used in practice to extract spatial features of data, but it often ignores the temporal correlation of data [44], while recurrent neural network and its variants have advantages in processing temporal data. Therefore, in order to extract features with spatial and temporal correlations from a given EEG signal, this paper combines both 1D-CNN and GRU networks. In addition, to improve the classification performance, this study also introduces an attention mechanism to form a hybrid network together with a 1D-CNN and GRU, and names the entire network model as the 1D-CNN-GRU-ATTN model.

The basic idea of building a hybrid neural network model is to combine two deep learning models, 1D-CNN and GRU, with their own advantages in series, where 1D-CNN is the first sub-network of the hybrid network, and GRU is the second sub-network of the hybrid network. Then, an attention mechanism, a fully connected layer, and an output layer are added behind the GRU network to form a hybrid network.

Subnet I in the hybrid neural network: 1D-CNN consists of two convolutional layers and one pooling layer. The convolutional layer is used to learn the high-level features of the EEG, and the size and number of convolution kernels are determined by the grid search method. The pooling layer selects max pooling to reduce the number of trainable parameters, and the size is determined by a grid search. Sub-network II: GRU is a gated recurrent unit, in which the number of hidden units is also determined by the grid search method, and then an attention mechanism is connected to screen the feature allocation weights output by the previous layer of the network. This is followed by a fully connected layer that combines all relevant features that are assigned more weights. The last layer is a classifier function, which maps the output of the neurons in the previous layer to the (0, 1) interval as the probability of each category, so as to achieve binary classification or multi-classification.

The detailed design of 1DCNN-GRU-ATTN is shown in Figure 5. After extracting the PSD of each band, the features are input into the convolution layer to extract the depth features, then the max pooling layer to reduce the network size, and then they are input into the convolution layer to continue the feature extraction. Then, the output of the CNN layer is used as the input of the GRU to extract the long-term dependence features of the signals. Next, the attention layer assigns different weights to the features extracted by the network to make the network focus on the features with larger weights, and selects more representative features from a large number of features to reduce the amount of calculation of the network. Then, the Fully Connected (FC) layer maps the feature space to the sample tag space. Finally, softmax implements classification.

In this study, the hybrid network uses the most common activation function, the rectified linear unit (ReLU), as the activation function of the hidden layer. Compared with other activation functions, ReLU has more efficient gradient descent and reverse propagation, avoiding the problems of gradient explosion and gradient disappearance [45]. In addition, the calculation process can be simplified. The expression of ReLU is as follows:(12)$$ReLU\left(x\right)=\{\begin{array}{c} x\;\;\;if\;x>0\\ 0\;\;\;if\;x\leq 0 \end{array}$$

In addition, in order to avoid over-fitting, the network also introduces dropout. In the network model training phase, dropout will randomly delete nodes from the hidden layer, and delete different nodes in each iteration, thereby effectively training different networks. The network chooses softmax as the classifier, classifies the output of the GRU, and displays the classification result in the form of probability.

In classification problems, especially when neural networks are used for classification problems, cross entropy is often used as a loss function. In addition, since cross entropy involves calculating the probability of each category, cross entropy appears in classification tasks with the sigmoid (or softmax) function almost every time. Therefore, this study adopts the cross entropy function as the loss function of the model. The cross entropy function is defined as follows:(13)$$loss=-\frac{1}{m}\overset{m}{\underset{j=1}{\sum }}\overset{n}{\underset{i=1}{\sum }}y_{ji}\log{\hat{y}}_{ji}$$
where *m* represents the number of samples, *n* represents the category, and $y_{ji}$ represents the true probability of the category, which represents the predicted probability.

The optimizer guides the parameters in the loss function to update in the correct direction during the back-propagation process of the neural network to ensure that the updated parameters can keep the loss function value close to the global minimum. The optimizer is an important part of model training. Because the RMSProp algorithm declines quickly and can adjust the learning rate adaptively, this paper chooses RMSProp as the model optimizer, in which the learning rate is set to 0.005.

This research uses the grid search method to adjust and optimize network parameters and hyperparameters.

(1)Network parameter tuning

For testing, 3000 samples of each record from the MODMA dataset are randomly selected. First, the epoch and batch size are set to 10 and 200, respectively. Then, the number of one filter in the convolutional layer, the number of two filters in the convolutional layer, the size of the convolution kernel, and the number of neurons in the GRU are defined as NFC_1, NFC_2, KSC, and NNG, respectively, for tuning. The grid search results are shown in Table 2. It can be seen that the model numbered M18 has the best performance.

(2)Hyperparameter tuning

The hyperparameters epoch and batch size are essential parameters in the neural network model, which are crucial to the classification results. After network parameter tuning, the same method is used to find the best combination of hyperparameters. According to Table 2, the M18, M21, and M24 models all show similar high accuracy. Therefore, the hyperparameters are optimized through experiments based on the different network parameters of the three models. It can be seen from Table 3 that when epoch and batch size are 100 and 256, respectively, the model produces the best performance.

## 3. Results

### 3.1. Diagnosis Process

The process of depression diagnosis using the hybrid neural network is shown in Figure 6.

The diagnosis process can be divided into the following five steps:
Step 1:Collecting and labeling the EEG signals of different subjects to form a dataset.Step 2:Preprocessing and extracting features of EEG signals.Step 3:Dividing the dataset into the training set, validation set, and test set to evaluate the performance of the depression diagnosis model.Step 4:Training the network model and saving it.Step 5:Verifying the effectiveness and sensitivity of the algorithm. The performance of the depression diagnosis model is evaluated based on the predicted and true labels.

### 3.2. Evaluation Metrics

In the classification task, we usually select the following performance indicators to evaluate the quality of the model.

(1)Accuracy [46] is defined as the ratio of the number of correctly classified samples to the total sample for a given test dataset.
(14)$$Accuracy=\frac{\left|TP\right|+\left|TN\right|}{\left|TP\right|+\left|FP\right|+\left|TN\right|+\left|FN\right|}$$

However, in the case of unbalanced positive and negative samples, this indicator has major flaws. For example, a set of test samples has a total of 1000, of which there are 900 positive samples and only 100 negative samples. Even if the classification model predicts all samples as positive, the accuracy is 90%. However, this indicator itself is not convincing because it cannot fully evaluate models.


(2)Precision [47] is the ratio of the number of correctly classified positive samples to the number of classified positive samples.
(15)$$Precision=\frac{\left|TP\right|}{\left|TP\right|+\left|FP\right|}$$(3)Recall [47] is the ratio of the number of correctly classified positive samples to the actual number of positive samples.
(16)$$Recall=\frac{\left|TP\right|}{\left|TP\right|+\left|FN\right|}$$(4)F1 score [48] is to evaluate the pros and cons of different algorithms. On the basis of precision and recall, the concept of *F*1 value is proposed to evaluate precision and recall as a whole.
(17)$$F1-score=2*\frac{Precision*Recall}{Precision+Recall}$$
where *TP*, *FP*, *TN*, and *FN* represent true positive, false positive, true negative, and false negative, respectively. True positives represent the number of samples predicted to be depression that are actually depression. False positives represent the number of samples predicted to be depressed that are actually healthy. True negatives represent the number of samples predicted to be healthy that are actually healthy. False negatives represent the number of samples predicted to be healthy that are actually depressed.


### 3.3. Experimental Result

The sample distribution of the public dataset and the private dataset is shown in Table 4.

(1)Experimental results and analysis of the MODMA dataset

When training the model, the samples in the dataset are first divided into the training set and test set according to the proportion of 9:1, and then one-tenth is randomly selected from the training set as the verification set. Therefore, the ratio of the training set, verification set, and test set is set to 0.81:0.09:0.1, respectively. The parameters of the model in the diagnosis of depression are shown in Table 5.

Figure 7 is an iterative curve of the 1D-CNN-GRU-ATTN model in the training process. The blue dotted line represents the accuracy of the training data, the blue solid line represents the accuracy of the verification data, the red dotted line represents the loss of training data, and the red solid line represents the loss of verification data. It can be seen that the hybrid model performs well on the public dataset, with high accuracy and low loss. In addition, the model can converge quickly without over-fitting. It shows that the model has a good generalization ability.

A confusion matrix is an effective visualization tool classification method for performance. Each row in the confusion matrix represents the basic fact, and each column represents the predicted label. The confusion matrix records the classification results on the test set. This model was tested on 1590 samples. The confusion matrix is shown in Figure 8, and the accuracy is 99.33%.

In addition, Table 6 also lists the classification report of the public dataset on the 1D-CNN-GRU-ATTN model. It can be seen that the precision, recall, and F1-score of the proposed model are close to or equal to 1, which proves that the method can effectively identify depression with a high classification accuracy and good generalization ability.

In order to further evaluate the performance of the proposed 1D-CNN-GRU-ATTN model, several groups of comparative experiments were carried out on the same public dataset. The comparison models used were 1D-CNN, LSTM, GRU, 1D-CNN-LSTM, and 1D-CNN-GRU. The parameters of these models are shown in Table 7. The values of epoch and batch size were set to the same as those of the 1D-CNN-GRU-ATTN model, which are 100 and 256, respectively.

Figure 9 shows the training curves of the six models on the public dataset, in which the solid line represents the accuracy and the dotted line represents the loss. It can be seen that the training effect of the hybrid network with the attention mechanism proposed in this study is better than that of the other five traditional neural networks.

In addition, the accuracy, loss, and training time of several different deep learning algorithms on public datasets are compared, as shown in Figure 10.

(2)Experimental results and analysis of the private dataset

Figure 11 is the training curve of the proposed 1D-CNN-GRU-GRU model and the comparison models (1D-CNN, LSTM, GRU, 1D-CNN-LSTM, and 1D-CNN-GRU) on the private dataset. The solid line represents the accuracy and the dotted line represents the loss. It can be seen that the accuracy of the model proposed in this study is significantly higher than that of the other models, and the loss is the lowest.

Similarly, the proposed 1D-CNN-GRU-ATTN model and the five mainstream models (1D-CNN, LSTM, GRU, 1D-CNN-LSTM and 1D-CNN-GRU) were compared in the experiment. The accuracy, loss, and training time of the private dataset on each model are shown in Table 8.

Figure 12 shows the precision, recall, and F1-score of the six models on the test set of the private dataset. It can be seen from the figure that the three evaluation indexes of the 1D-CNN-GRU-ATTN model are higher than those of other comparable models.

Compared with the public dataset MODMA, the amount of private data samples is smaller. By comparing the evaluation metrics of different algorithms in the figure, it can be seen that the performance of the method proposed in this paper is better.

Table 9 compares the results of several different methods for depression diagnosis on public datasets. Sun et al. [49] extracted different types of EEG features, including nonlinear and functional connectivity features (phase lag index, PLI), comprehensively analyzing the EEG signals of major depressive patients. Although the nonlinear and PLI features have a good explanatory power, the computation is complex, and the classification effect using the machine learning classifier is not good. Wang et al. [50] used an alternative time-frequency analysis technique based on intrinsic time-scale decomposition (ITD) with TCN and L-TCN and obtained a better result than the literature [49]. Compared with the above features, the PSD feature combined with the hybrid neural network has a better prediction effect on depression, and its accuracy is higher than that of other studies. The 1D-CNN-GRU-ATTN model proposed in this paper has achieved the better classification results, with the highest classification accuracy of 99.33%.

## 4. Discussion

In this study, a high-performance hybrid neural network depression detection method is proposed based on deep learning technology. Different from the previous studies, we concatenated 1D-CNN and GRU, and introduced the attention mechanism to form the hybrid neural network. Figure 8 shows that the model has the highest accuracy on the public dataset, and the evaluation metrics in Table 6 are close to one, which indicates that the model has an outstanding generalization ability. Table 8 and Figure 12 show that the model is not only applicable to the public dataset, but also to the private dataset. The evaluation metrics are higher than those of 1D-CNN, LSTM, GRU, 1D-CNN-LSTM, and 1D-CNN-GRU. In terms of training time, 1D-CNN has the shortest training time compared to the others because of its simple structure. The gate structure of GRU is simpler than LSTM, which leads to a slightly faster training speed than LSTM. To summarize, the hybrid network proposed in this study has a higher accuracy and shorter training time than other comparison algorithms. Figure 11 can better reflect the advantages of the 1D-CNN-GRU-ATTN model than Figure 9. The public dataset is “almost clean and perfect”, and all models have a classification accuracy higher than 90%. The curves in Figure 9 are relatively close. On the private dataset, the 1D-CNN-GRU-ATTN model shows outstanding advantages.

The proposed method has several advantages. First, it can achieve a higher classification accuracy than the existing methods for classification between depression patients and normal subjects based on the same dataset. Second, the network model introduces the attention mechanism, which effectively reduces the training time. According to the results, it can be explained that the attention mechanism focuses computing resources on features with large weights, thereby reducing time overhead. Experimental results verify that the hybrid model proposed in this paper achieves a high performance.

## 5. Conclusions

This research took EEG signals as the research object, the public dataset and private dataset as the research basis, and used deep learning algorithms as the research theory to deeply study the characteristics of EEG signals and propose a high-performance mental state detection method. First, the network parameters and hyperparameters of the model were optimized through experiments, and then the effectiveness and advancement of the method were verified through comparative experiments. The experimental results show that: (1) The proposed method in this paper is more effective in the diagnosis and recognition of depression. (2) In the case of a small number of iterations, this method can adaptively extract the features of the EEG signals to achieve a higher classification accuracy than other methods. (3) This method has excellent performance on public and private datasets and provides a technical basis for the diagnosis and screening of depression. The effectiveness and advantages of the method were verified through comparative experiments. The method proposed in this paper had an accuracy of 99.33% on the public dataset and a satisfactory accuracy of 97.98% on the private dataset.

Although the improved hybrid network model achieved good results in mental state detection, there are still some problems to be improved, as follows: (1) The dataset used in this study can meet the research requirements in this paper; however, there were an insufficient number of negative samples. We hope to collect more clinical data in the future to improve the generalization ability of the model. In addition, this study only focused on the detecting the depression state without treatment, and nondestructive therapy research can be considered for future work. (2) In this study, when collecting EEG signals, the subjects were equipped with heavy electrode caps which needed to be pressed to make them fully contact with the scalp surface, which may have caused discomfort for some users. In future research, we can consider using fewer and lighter electrodes with portable acquisition devices, such as ear-BCI. (3) In this study, only qualitative analysis was carried out on the psychological state, and no quantitative analysis. Quantitative analysis of the psychological state can be added to the future research. For example, depression can be detected according to the degree, which can be divided into normal, mild, moderate, and severe; the attention mechanism can be set as an indicator to indicate the degree of concentration; etc.

## Figures and Tables

**Figure 1 brainsci-12-00834-f001:**
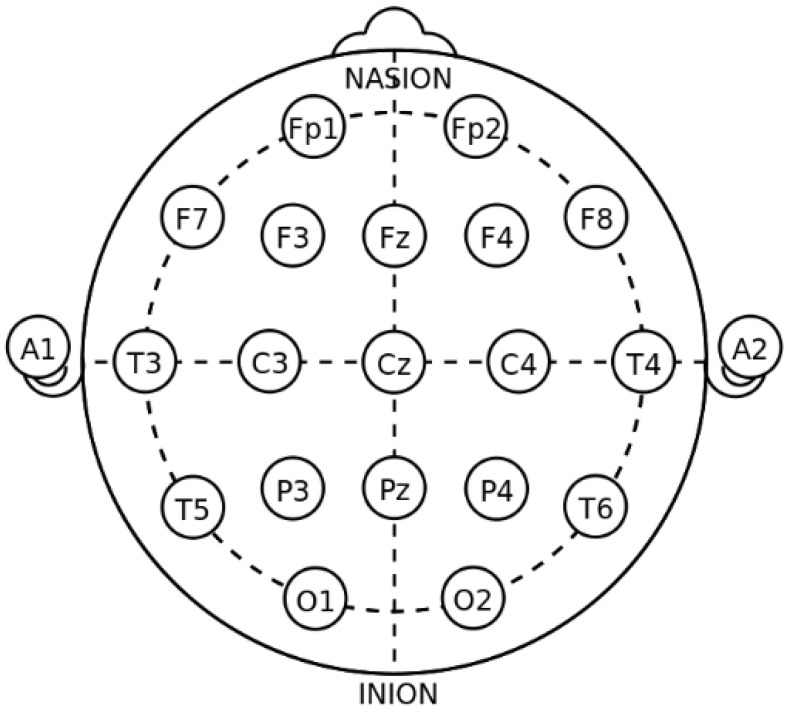
Position of 16 electrodes in international 10–20 system.

**Figure 2 brainsci-12-00834-f002:**
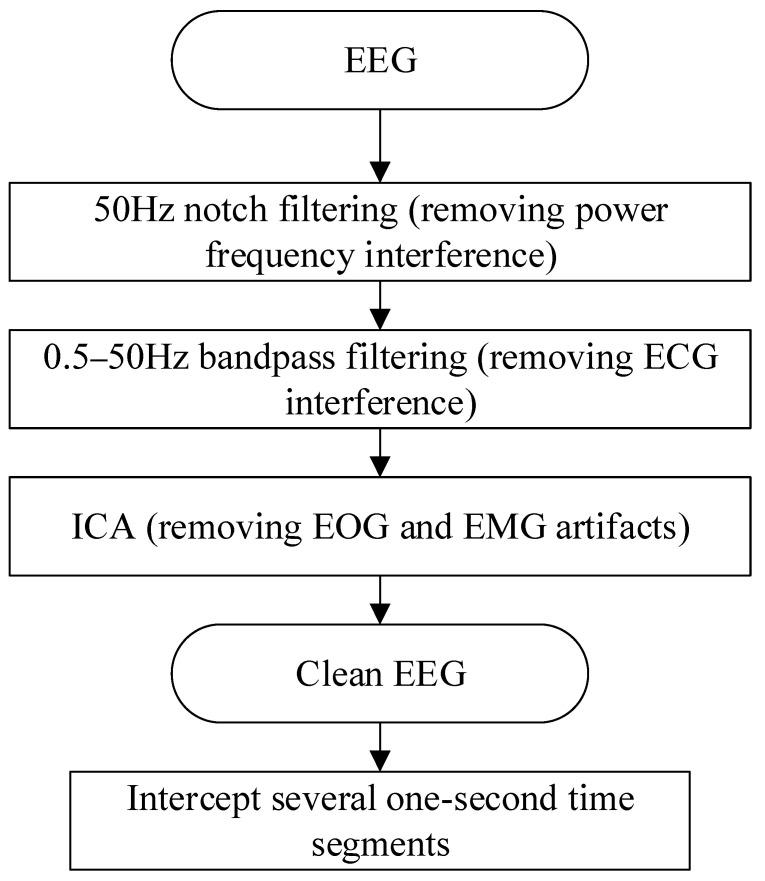
EEG preprocessing process.

**Figure 3 brainsci-12-00834-f003:**
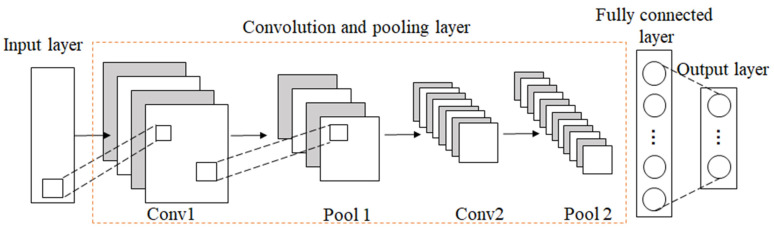
The architecture of CNN.

**Figure 4 brainsci-12-00834-f004:**
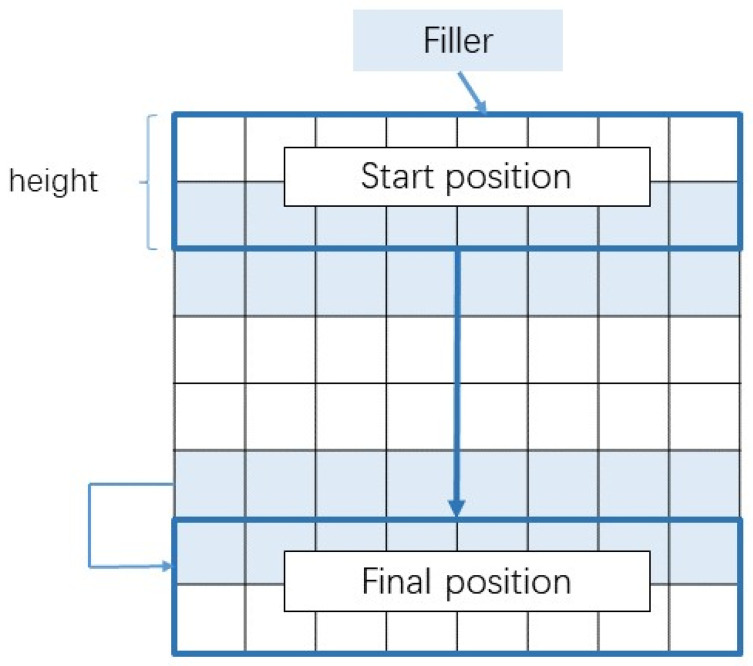
The sliding direction of the filter in 1D-CNN.

**Figure 5 brainsci-12-00834-f005:**
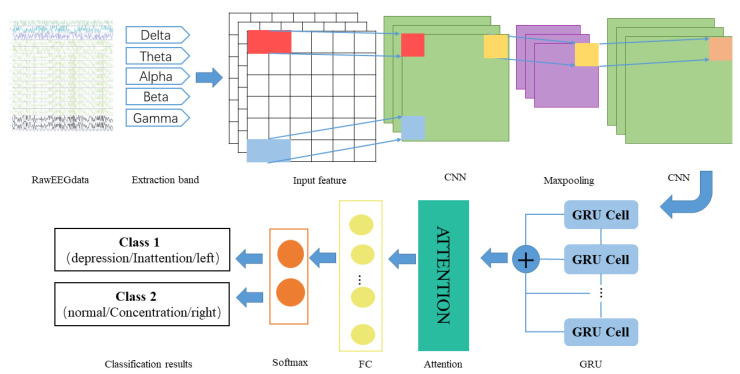
The structure of the hybrid network.

**Figure 6 brainsci-12-00834-f006:**
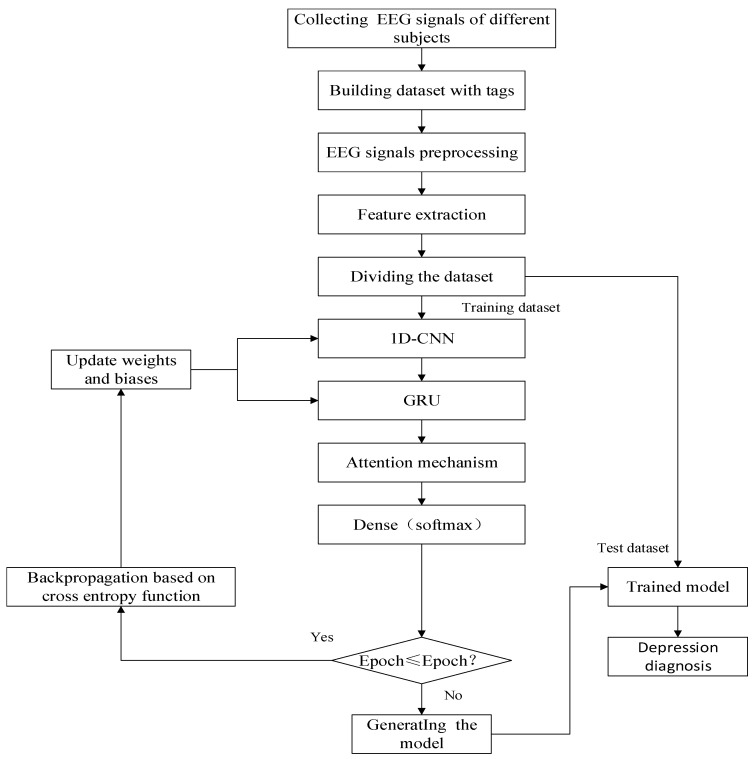
Depression diagnosis algorithm flow.

**Figure 7 brainsci-12-00834-f007:**
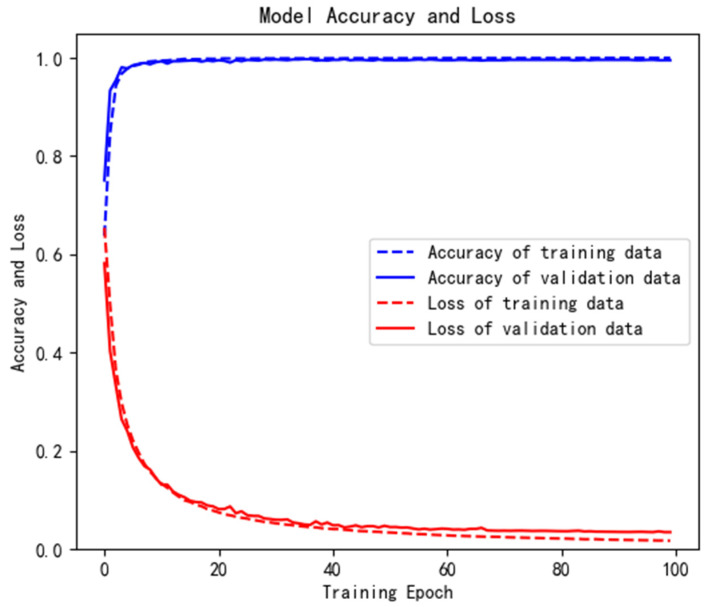
Iterative curve of training process.

**Figure 8 brainsci-12-00834-f008:**
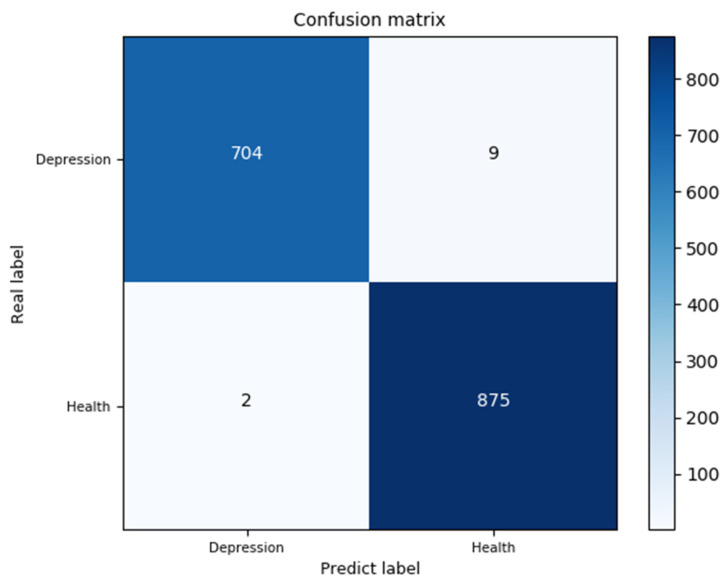
Confusion matrix of the test set.

**Figure 9 brainsci-12-00834-f009:**
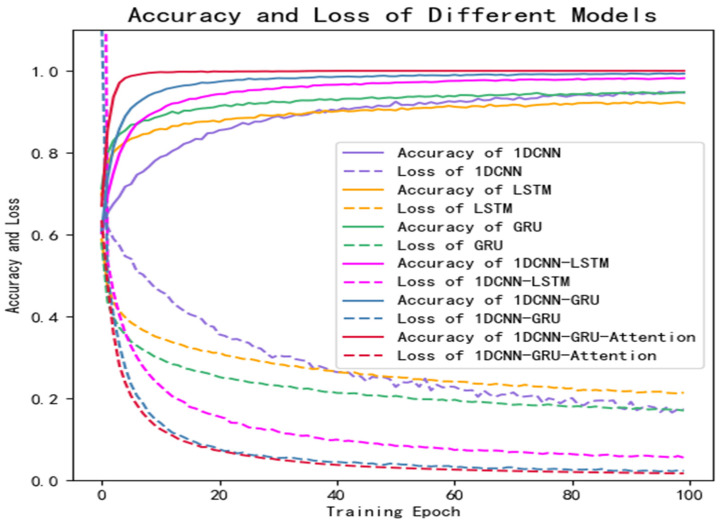
Results of comparative experiments on public dataset.

**Figure 10 brainsci-12-00834-f010:**
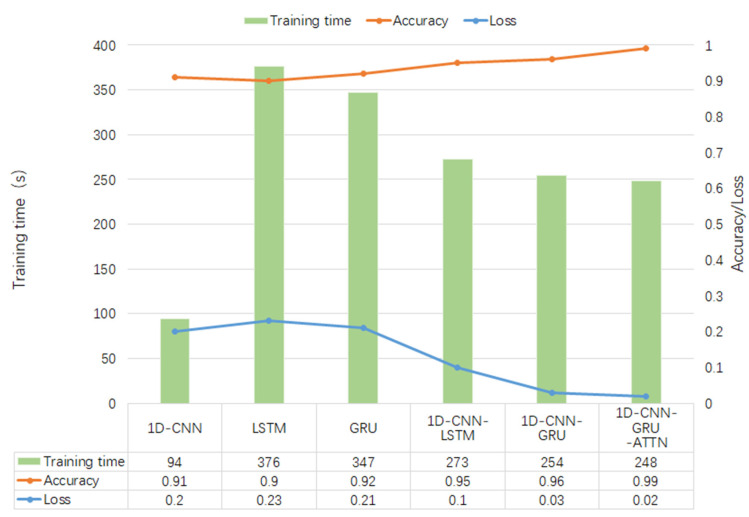
Results of comparative experiments on public dataset.

**Figure 11 brainsci-12-00834-f011:**
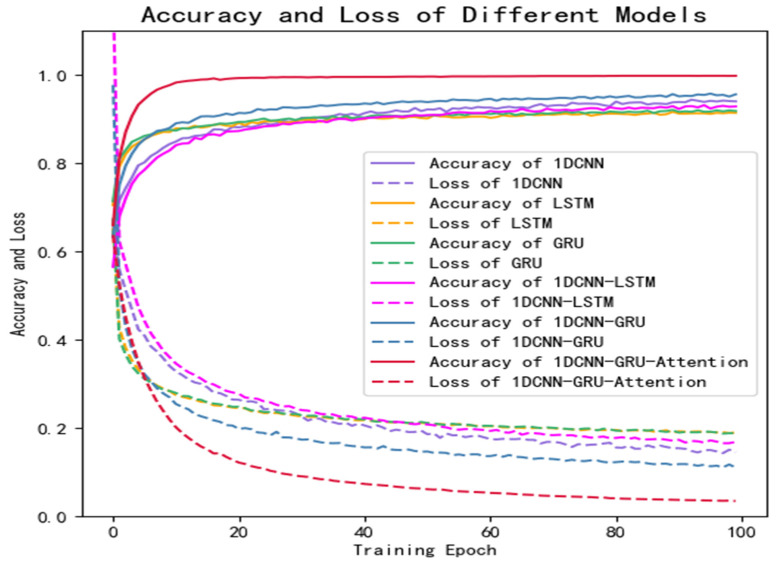
Results of comparative experiments on private dataset.

**Figure 12 brainsci-12-00834-f012:**
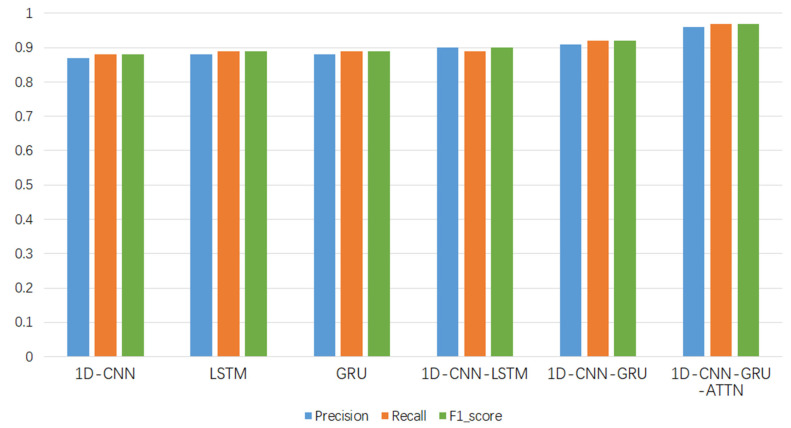
Evaluation indicators of each model.

**Table 1 brainsci-12-00834-t001:** Properties of datasets with data.

Dataset	Properties	Values
Public Dataset (MODMA)	Number of subjects	53
	Number of subjects with depression	24
	Male/female ratio	33/20
	Number of channels	128
	Sampling rate (Hz)	250
Private Dataset	Number of subjects	32
	Number of subjects with depression	16
	Male/female ratio	16/16
	Number of channels	16
	Sampling rate (Hz)	250

**Table 2 brainsci-12-00834-t002:** Tuning of network parameters.

Model	NFC_1	NFC_2	KSC	NNG	ACC	Loss
M1	128	128	3	64	0.75	0.38
M2	128	128	3	128	0.75	0.31
M3	128	128	3	256	0.79	0.30
M4	128	256	3	64	0.77	0.31
M5	128	256	3	128	0.77	0.32
M6	128	256	3	256	0.75	0.29
M7	256	128	3	64	0.76	0.34
M8	256	128	3	128	0.75	0.33
M9	256	128	3	256	0.75	0.38
M10	256	256	3	64	0.74	0.36
M11	256	256	3	128	0.75	0.35
M12	256	256	3	256	0.75	0.28
M13	128	128	5	64	0.87	0.21
M14	128	128	5	128	0.88	0.20
M15	128	128	5	256	0.89	0.19
M16	128	256	5	64	0.90	0.27
M17	128	256	5	128	0.89	0.17
M18	128	256	5	256	0.92	0.16
M19	256	128	5	64	0.89	0.23
M20	256	128	5	128	0.89	0.17
M21	256	128	5	256	0.91	0.19
M22	256	256	5	64	0.89	0.21
M23	256	256	5	128	0.90	0.18
M24	256	256	5	256	0.91	0.17

**Table 3 brainsci-12-00834-t003:** Tuning of hyperparameters.

Network Parameters	Model	Epoch	Batch Size	Accuracy
M18	M1	80	128	0.9776
	M2	90	128	0.9760
	M3	100	128	0.9770
	M4	80	256	0.9780
	M5	90	256	0.9753
	M6	100	256	0.9797
M21	M7	80	128	0.9530
	M8	90	128	0.9532
	M9	100	128	0.9541
	M10	80	256	0.9523
	M11	90	256	0.9537
	M12	100	256	0.9548
M24	M13	80	128	0.9533
	M14	90	128	0.9531
	M15	100	128	0.9536
	M16	80	256	0.9540
	M17	90	256	0.9542
	M18	100	256	0.9547

**Table 4 brainsci-12-00834-t004:** The sample distribution of public dataset and private dataset.

Dataset	Depression	Normal	Samples
MODMA	7200	8700	15,900
Private dataset	4119	3114	7533

**Table 5 brainsci-12-00834-t005:** Parameters in the diagnosis model of depression.

Description	Value
Number of filters in convolutional layer 1	128
Number of filters in convolutional layer 2	256
Filter size in convolutional layers 1 and 2	5
Pooling size in the max pooling layer	2
Number of neurons in GRU	256
Dropout	0.2
Epoch	100
Batch size	256

**Table 6 brainsci-12-00834-t006:** Classification report.

Label	Description	Precision	Recall	F1-Score	Support
0	Depression	1.00	0.99	0.99	713
1	Health	0.99	1.00	0.99	877

**Table 7 brainsci-12-00834-t007:** Parameters of comparison models.

1D-CNN	LSTM	GRU	1D-CNN-LSTM	1D-CNN-GRU
Conv1–5×128 Maxpool-2 Conv2–5 × 256 Maxpool-2 FullyConnected-256 Softmax-2	LSTMcell-256 FullyConnected-64 Softmax-2	GRUcell-256 FullyConnected-64 Softmax-2	Conv1–5 × 128 Maxpool-2 Conv2–5 × 256 LSTMcell-256 Softmax-2	Conv1–5 × 128 Maxpool-2 Conv2–5 × 256 GRUcell-256 Softmax-2

**Table 8 brainsci-12-00834-t008:** The accuracy, loss, and time of private dataset on different models.

Model	Accuracy (%)	Loss	Train Time (s)
CNN	86.38	0.33	64.38
LSTM	88.22	0.30	253.62
GRU	88.65	0.30	227.43
1D-CNN-LSTM	90.08	0.27	178.32
1D-CNN-GRU	91.43	0.26	164.57
1D-CNN-GRU-ATTN	97.98	0.07	160.48

**Table 9 brainsci-12-00834-t009:** The comparison of the classification results between the proposed method and previous works.

Author	Year	Features	Methods	Accuracy (%)
Shuting, S. et al. [49]	2020	Nonlinear + PLI	LR + ReliefF	81.79
Wang, Y. et al. [50]	2021	ITD + statistical features	TCN	85.23
Wang, Y. et al. [50]	2021	ITD + statistical features	L-TCN	86.87
This paper	2022	PSD of 5 bands	1D-CNN-GRU-ATTN	99.33

## Data Availability

Publicly available datasets were analyzed in this study. The data can be found here: http://modma.lzu.edu.cn (accessed on 10 May 2021). The data presented in this study are available upon request from the corresponding author.

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
