# Peer review of "A Depression Diagnosis Method Based on the Hybrid Neural Network and Attention Mechanism"

_brainsci, 2022, doi:10.3390/brainsci12070834_

Round 1

Reviewer 1 Report

Dear authors,

I present my suggestions and evaluation of your manuscript “Depression diagnosis method based on neural network and attention mechanism” below.  I consider the topic of machine learning in the case of biological signal classification to be current. And the topic of depression and its diagnosis (or consequently trying to identify the response to antidepressants) is a really interesting and popular topic. 

In general the main aim of the study should be highlighted and the state of the art part needs to be enlarged by recent literature.  The methodology part of the manuscript should be clarified. The discussion part is completely missing. I am missing the comparison to other recent studies. The parameters of the proposed network should be clarified and description of other classifiers need to be added. 

I am adding the main points and questions that need to be solved or clarified. The list of these points are divided into the major and minor ones.

Major

  1. The introduction did not provide sufficient background and did not include all relevant references.

  2. Private dataset should be described more precisely, the gender and age should be added. 

  3. The equations should be better set into the text. If they are the general one then they should be cited. Please, use better typos (for example when you described variables the text should not be offset, the variables should be in italic style and other text in equations should be in normal style).

  4. The data were filtered by an IIR filter. Why was the IIR type chosen? Standardly the FIR filter is used. Can you show the characteristic of this filter? Was it steep enough and what about ripples?

  5. There are some formulas that need to be clarified or referenced. For example:

- CNN is widely used in practice to extract spatial features of data, but it often ignores the temporal correlation of data.
- Compared with other activation functions, ReLU has more efficient gradient descent and reverse propagation, avoiding the problems of gradient explosion and gradient disappearance. 

  1. Authors mentioned the filtering because of artifacts. From the diagram on Figure 2 it is obvious that the 0.5-50 Hz filtering was done to avoid the EMG and ECG artifacts. But the EMG artifacts range from 20 Hz. It should be clarified which type of ICA was used for removing EOG artifacts (maybe add reference).

  1. It is not valid to extract the PSD to 50 Hz in case of sampling frequency 125 Hz, 128 Hz respectively, in case of EEG signal analysis. 

  2. Why was the rectangular window used in PSD estimation? Standard is to use a Hanning/Hamming window to avoid the power spectra leakage. What about overlapping of the segments? 

  3. The random selection of 3000 samples from MODMA was done on the patient level or some segments of each record were selected?

  4. Part (1) Network parameter tuning: How were the other parameters set during this optimization phase?

  5. The discussion part is completely missing. 

  6. The results of the proposed model are compared to some other classificators, but their parameters are completely missing. Please, describe the 1D-CNN, LSTM, GRU, 1D-CNN-LSTM an 1D-CNN-GRU by its parameters. 

  7. The comparison with similar research studies is missing. Please compare this method to the current one methods.

Minor

  1. I recommend adding the authors’ ORCiD.

  2. I recommend publishing the codes as well (for example on github) based on the Open Science principles.

  3. The figures should be cited if they are taken over. For example Figure 1.

  4. Did You consider testing this method on the 128 electrodes system on the MODMA dataset? How can the number of electrodes influence the final topography of NN?

  5. The accuracy during the first tuning first phase of network parameter tuning is 0.91 in case of M21 and M24 which is really close to the best one, M18. Did you test the second part of optimisation for these two cases? This part should be discussed. 

  6. The resulting accuracy has different decimal numbers in case of the first and second part of hyperparameter optimization (Table 2 and 3). Why? It seems that in Table 3, the resulting accuracy is close to each other across the models. 

  7. Please, clarify the ratio between the training:validation: testing datasets.

  8. Are the results of classification described for the testing part of the data only?

Reviewer 2 Report

The authors proposed an approach based on a hybrid neural network (1D-CNN-GRU-ATTN), using EEG signals as input, for the detection of depression. The method combines a one-dimensional convolutional neural network with a gated recurrent unit and introduces an attention mechanism. The method was tested on a public and a private dataset and its performance was compared with that of several deep learning algorithms. The results revealed that the 1D-CNN-GRU-ATTN model has the best performance for both datasets. The topic is interesting, the paper is well-structured and exhaustive. The methodology is explained in detail and the results are clearly presented. I have only a few suggestions:

1. In the abstract you wrote that the hybrid network has the best performance but you do not specify that other deep learning algorithms were tested. Add that and the numerical values of the hybrid neural network accuracy for both datasets.

2.      Some references are missing throughout the paper. Please add them (lines 48-56, 59-61, 66-68, 83-88, 120, 151-154, 197-199).

3.      Define all abbreviations before using them.

4.  Replace “electroencephalogram (EOG)” with “electrooculogram (EOG)” (line 138).

5.      Add a figure showing the architecture of a CNN (Section 2.2.1).

In my opinion, future work testing this method using more EEG channels could be interesting.

Reviewer 3 Report

The overall contents of this manuscript is not well organized to give a clear overview of this work. I have suggested some major comments and suggestions about this study are as the following:

Comments to the Authors:

  1. Authors should re-write the abstract of this study clearly with including objective, method, results, conclusion and significance.
  2. The introduction of this study is looking very weak. It should be divided into three paragraphs like background, gap between previous research to new develop method and hypothesis with specific aims.
  3. There is no Discussion section in this paper. Authors should write discussion clearly like how these results is consistent or inconsistent with previous research published in this field.
  4. Authors should refer some latest related papers in introduction and discussion sections, and the published articles based on attention. 
  1. My suggestion is that the authors should write some limitations of this model and future application.

Round 2

Reviewer 1 Report

Dear authors, I attached the file with response to my comments.
